# Different Disease Levels Reveal Kiwifruit Brown Spot Impacts on Fruit Yield and Quality

**DOI:** 10.3390/jof11080593

**Published:** 2025-08-15

**Authors:** Yuhang Zhu, Jing Xu, Jun Wang, Rui Yang, Wen Chen, Kaikai Yao, Miaomiao Ma, Qinghua Chen, Zhonghan Fan, Cuiping Wu, Rongping Hu, Guoshu Gong

**Affiliations:** 1Plant Protection Department, College of Agronomy, Sichuan Agricultural University, Chengdu 611130, China; s20163305@stu.sicau.edu.cn (Y.Z.); b20171803@stu.sicau.edu.cn (J.X.); wangjun1@stu.sicau.edu.cn (J.W.); 2020201073@stu.sicau.edu.cn (R.Y.); 2020301154@stu.sicau.edu.cn (W.C.); yaokaikai@stu.sicau.edu.cn (K.Y.); 14915@sicau.edu.cn (M.M.); 71262@sicau.edu.cn (C.W.); 2Institute of Plant Protection, Sichuan Academy of Agricultural Sciences, Chengdu 610066, China; fenglq2004@sina.cn (Q.C.); fzhmer@163.com (Z.F.)

**Keywords:** *Corynespora cassiicola*, yield loss, fruit quality, principal component analysis, damage threshold

## Abstract

Kiwifruit brown spot, caused by the fungus *Corynespora cassiicola*, has recently emerged as a problematic foliar disease of kiwifruit, causing premature defoliation. The objective of this study was to determine the effects of kiwifruit brown spot on the yield and quality of kiwifruit. Principal component analysis (PCA) was used to conduct a comprehensive evaluation of the fruit quality of ‘Hongyang’ kiwifruit in the main producing regions. The first principal component for PCA included the weight of individual fruit, soluble solids content, and dry matter content, which were negative significantly correlated with disease index. The significant differences among different disease levels indicated that the impact of the disease on fruit quality was largely determined by these three intrinsic flavor indices. Due to kiwifruit brown spot, the average yield loss was 22.652%, which leads to kiwifruit quality being downgraded by one grade, resulting in an economic loss of 73,591 yuan/ha. The Pearson correlation coefficient between disease index and comprehensive score of fruit quality was −0.762 (*p* < 0.01), indicating a significant relationship. Accordingly, the disease loss model was constructed, and the damage threshold based on disease index for kiwifruit brown spot was calculated to be 36.14. In conclusion, this study found that kiwifruit brown spot could have a significant impact on yield and fruit quality.

## 1. Introduction

Kiwifruit (*Actinidia chinensis* Planch) is one of the most delicious fruits worldwide, with its special flavor and abundant nutrients, minerals, and dietary fiber [1,2], and is grown mostly in China, Italy, and New Zealand [3,4]. China currently has the largest planting area and output of kiwifruit in the world [5]. The kiwifruit industry in Sichuan has experienced unprecedented development since 2008, with its current output ranking second in China. In recent years, kiwifruit diseases have become one of the most significant factors restricting industry development due to the focus on a single variety and the unreasonable layout of the planting area. With the continuous cultivation of the highly susceptible varieties, kiwifruit brown spot caused by *Corynespora cassiicola* [6,7] is widely distributed across the entire producing region in Sichuan [8], and is the second most important disease following kiwifruit bacterial canker disease caused by *Pseudomonas syringae* pv. *actinidae* [9]. Kiwifruit brown spot primarily causes premature defoliation, leading to premature sprouting of autumn shoots, and the fruits experience severe water loss, shrivel, and soften, losing value as a commodity and causing huge yield loss [9].

Although the leaf-spot disease does not directly cause the destruction of trees, it can cause severe leaf necrosis and even a large amount of fallen leaves, resulting in a reduced photosynthetic area, thereby seriously affecting the yield and quality of fruit [10,11,12]. Holcman et al. [13] revealed that the amount of damage inflicted on vine yield and grape quality were related to the occurrence of downy mildew caused by *Plasmopara viticola*. The globally expanding Alternaria leaf blotch in apple caused by the fungal genus *Alternaria* can lead to severe leaf necrosis, premature defoliation, and large economic losses [14]. *Corynespora cassiicola* has a wide range of hosts, which has an adverse impact on the yield and quality of other hosts. According to Hagan et al. [15], it was estimated that susceptible cultivars could lose up to 448 kg/ha of yield due to *C. cassiicola* cotton target spot. Bowen et al. [16] has developed a yield loss model for cotton target spot to estimate yield losses. Target leaf spot caused by *C. cassiicola* has posed a significant threat to greenhouse-cultivated cucumbers in China [17], and Corynespora Leaf Fall disease also had a substantial economic impact on rubber-producing nations [18].

The research on kiwifruit brown spot is still in the initial stage, and focuses primarily on pathogen identification, population diversity of the pathogen, cultivar resistance, and disease-control techniques [6,7,19,20]. The effect of the disease on the yield and quality of kiwifruit remained unclear, but other kiwifruit diseases have been revealed. For example, Wang et al. [21] determined that the titratable acidity and vitamin C (VC) content, firmness, lightness, and saturation decreased, whereas the soluble solids content (SSC) increased after infection with *Penicillium expansum*. Wu et al. [22] reported that *Actinidia* yellowing ringspot virus (AYRSpV) infection could reduce the chlorophyll content of Hayward kiwifruit by 74.61% to 76.64%, fruit yield by 14.50% to 24.10%, sugar-to-acid ratio by 50.09% to 50.57%, and fruit dry matter content by 1.67% to 1.78%. Although they have reported the effects of kiwifruit diseases on yield and quality using artificial methods, the effects under natural conditions have yet to be fully comprehended.

In the current study, the kiwifruit quality of three main producing regions in Sichuan was measured between 2019 and 2021. The correlation between the severity of kiwifruit brown spot and each quality index was analyzed, and a comprehensive evaluation was carried out by principal component analysis. The yield loss model was constructed, and the damage threshold of the disease was determined. The objectives of this study were not only to quantify yield loss due to premature defoliation caused by *C. cassiicola*, but also to reveal the effects of kiwifruit brown spot on the quality of kiwifruit, aiming to provide reasonable guidance on the field prevention and control of kiwifruit brown spot.

## 2. Materials and Methods

### 2.1. Design of Field Experiment

The study was initiated in 2019 in the main kiwifruit-producing regions, located in Anzhou, Dujiangyan, and Lushan, and was repeated in 2020 and 2021. The experimental cultivar ‘Hongyang’ is extremely susceptible to kiwifruit brown spot [19]. Representative experimental sites were selected in each region, with an area of about one hectare per site, a tree age of over five years, flat terrain, and even tree growth. The site conditions of the three regions were mountain yellow-brown soil in Anzhou, plain brownish-yellow sandy loam soil in Dujiangyan, and mountain red soil in Lushan, respectively.

To clarify the effect of different severity of kiwifruit brown spot on yield and fruit quality, 42.4% fluzolamide·pyraclostrobin SC (BASF Corp., Ludwigshafen, Germany) at 480 mL/ha was applied 0 times, 1 time (only applied on June 25), and 3 times to establish three experimental treatments with three disease levels. Field treatments were arranged in a randomized block design, and each treatment consisted of five replicated plots, with an area of about 200 m^2^ per plot, resulting in a total of fifteen plots in each site. The same field management was performed in the same experimental site during the entire growing period except for the fungicide treatments. The fungicide applications were initiated at the onset of disease on 25 June, with subsequent applied for 15-day intervals. Some plots were finally excluded due to some uncontrollable factors in the field, such as out-of-control fungicide effects (continuous rainfall), fruit rot [23], root rot [24], drought, waterlogging, etc. Thirty, twenty-five, and twenty-six plots which could represent different disease levels were included in Anzhou, Dujiangyan, and Lushan, respectively, totaling eighty-one plots over three years.

### 2.2. Measurement of Disease Severity

Disease investigation and fruit harvesting were conducted 20 days (29 August) after the third fungicide application. Each tree was examined from five different directions (east, west, south, north, and middle), with two fruiting branches randomly selected in per direction. From each branch, five leaves were selected, starting from the base upwards, resulting in a total of fifty leaves per tree. Disease severity was rated using a 0 to 9 leaf-spot-modified grading system [20]: 0 = no visible lesion; 1 = disease spot of leaf area <5%; 3 = 5 to 25%; 5 = 26 to 50%; 7 = 51 to 75%; 9 = more than 75% and defoliation (Figure 1). Disease index (DI) was calculated according to the following formula. Disease level: low (DI < 33.33), medium (33.33 ≤ DI < 66.67), and high (DI ≥ 66.67) [8].
$$DI=\frac{\sum \left(the\;{leaves}^{’}\;number\;of\;every\;grade\times the\;value\;of\;disease\;grade\right)}{the\;number\;of\;all\;leaves\times the\;value\;of\;highest\;grade}\times 100$$

### 2.3. Assessment of Kiwifruit Quality

The fruit of the corresponding plant was collected while the disease was investigated. A total of 138 kiwifruits from each plot were chosen without any mechanical damage or disease. On the day of harvest, the harvest indices (shape index, weight of individual fruit, firmness, and soluble solids content) of 60 kiwifruits from each plot were determined. Eighteen kiwifruits were collected to determine dry matter content, with repetitions carried out three times for each group with six kiwifruits. Sixty kiwifruits were stored at room temperature, and postharvest indices (firmness, soluble solids content, shelf life, and incidence of fruit rot) were measured after natural soft ripening.

The weight of individual fruit was determined using an electronic balance (ATY224, SHIMADZU, Kyoto, Japan). Equatorial and longitudinal diameters of the fruit were measured using vernier calipers (MNT101802, Meinaite, Berlin, Germany). Kiwifruit firmness was measured on two points at equal distances along the equator of each fruit with a penetrometer (GY-4, Fangke Ltd., Weifang, China). Soluble solids content of kiwifruit juice was assessed with refractometer (MIK-B55, MEACON Ltd., Hangzhou, China). For the determination of dry matter content, each kiwifruit was cut into thin slices and the fresh weight of each fruit was measured. Then, the slices were put into the oven at 60 °C. After 60 h, the dry weight of the fruits was determined until the slices were completely dried. Shape index was the ratio of the longitudinal diameter to the equatorial diameter, and dry matter content was the ratio of dry weight to fresh weight. The coefficient of variation (*CV*) of each fruit index was calculated.
Coefficient variation=Standard deviation/Mean×100%

### 2.4. Estimate of Yield Losses

The yield and grade of kiwifruit were primarily determined by the weight of individual fruit (*WIF*). The effect of disease on yield was evaluated by regressing between weight of individual fruit and disease index estimated in each site. The calculated linear model was evaluated for fit by examining coefficients of determination (R^2^) and significance level (*p*). The theoretical value and theoretical grade correspond to the value and grade with no disease occurrence, respectively. The following described the kiwifruit grade standards.

Grade: first (I): *WIF* ≥ 120 g, 24 yuan/kg; second (II): 100 g ≤ *WIF* < 120 g, 20 yuan/kg; third (III): 80 g ≤ *WIF* < 100 g, 16 yuan/kg; fourth (IV): 60 g ≤ *WIF* < 80 g, 12 yuan/kg; fifth (V): *WIF* ≤ 60 g, 8 yuan/kg.
$$YL(\%)=\frac{theoretical\;value\;of\;WIF-average\;value\;of\;WIF}{theoretical\;value\;of\;WIF}\times 100\%$$

### 2.5. Principal Component Analysis (PCA)

The entire annual dataset was standardized using the membership function method prior to PCA: Positive correlation index (shape index, weight of individual fruit, firmness, soluble solids content, dry matter content, and shelf life) was calculated according to Equation (1), while the negative correlation index (disease incidence of fruits) was calculated according to Equation (2). PCA was performed with SPSS 21.0, and the first few principal components with eigenvalues greater than 1.0 and a cumulative contribution rate of variance greater than 85% were utilized for further analysis [25]. The comprehensive scores (*D_n_*) of each plot were obtained by Equation (3).
(1)$$U_{in}=\frac{X_{in}-X_{imin}}{X_{imax}-X_{imin}}$$
(2)$$U_{in}=1-\frac{X_{in}-X_{imin}}{X_{imax}-X_{imin}}$$
(3)$$D_{n}=\sum_{j=1}^{m}F_{jn}\times E_{j}$$ where
$U_{in}$ refers to the membership function value of the ith quality index raw data of the NTH sample after transformation.
$X_{in}$ refers to the original determination data of the ITH quality index of the NTH sample;
$X_{imax}$ and
$X_{imin}$ refer to the maximum and minimum values of the raw data of the ith quality index in all samples, respectively.
$D_{n}$ is the comprehensive score of fruit quality of each plot obtained by PCA;
$F_{jn}$ refers to the JTH principal component score of NTH samples; M refers to the number of extracted principal components;
$E_{j}$ refers to the contribution rate of JTH principal component.

### 2.6. Statistical Analysis

Experimental data were analyzed with SPSS 21.0 (SPSS Inc., Chicago, IL, USA). The significance of differences between treatments were determined at the 5% level using ANOVA followed by Duncan’s new multiple-range method. The correlation between each index was analyzed using Pearson correlation. The plotting of the heat map and the screening and construction of the optimum model were performed using Origin 2019b (OriginLab, Northampton, MA, USA).

## 3. Results

### 3.1. Model Screened and Constructed

During the determination of fruit-quality indices, there was a significant correlation between firmness and soluble solids content, with a Pearson correlation coefficient of −0.843 (*p* < 0.01). To further examine their correlation, common mathematical models were screened. As shown in Table 1 (only the optimum model for each function was presented), the Boltzmann model could reveal this. The corresponding equation was developed based on the firmness and soluble solids content of each plot (Appendix A), and the value corresponding to a firmness of 1 (the edible standard) was uniformly taken as the value of soluble solids content.

### 3.2. Effects of Disease Severity on Fruit Quality of Kiwifruit

Disease index and the fruit quality was measured between 2019 and 2021 (Table 2). The descriptive statistical results showed that disease index (DI) of 81 plots (*n* = 81) conformed to a normal distribution (Appendix A), indicating there was statistical significance among three disease levels (high, medium, and low). Among the fruit-quality indices (Table 2), the average value of weight of individual fruit (WIF), shape index (SI), firmness (FM), soluble solids content (SSC), dry matter content (DMC), disease incidence of fruit rot (DIF), and shelf life (SL) were 94.18 ± 20.89 g, 1.102 ± 0.068, 5.550 ± 0.826 kg/cm^2^, 16.77 ± 1.89%, 19.13 ± 1.81%, 3.64 ± 7.25%, and 15.49 ± 6.88 d, respectively. The majority of the fruit-quality data conformed to a normal distribution (Appendix A), with the exception of incidence of fruit rot (*p* = 0.005 < 0.05). Coefficient variation of incidence of fruit rot was the highest (197.73%), shape index was the lowest (6.11%), and other fruit-quality indices ranged from 9.39% to 44.13% (Table 2). Each fruit-quality index exhibited distinct characteristics, so it is necessary to evaluate them comprehensively.

The correlation between the occurrence of kiwifruit brown spot and each fruit-quality index was revealed (Figure 2). The result indicated a significant correlation between disease index and weight of individual fruit, with the Pearson correlation coefficient ranging from −0.651 to −0.988. As with weight of individual fruit, soluble solids content and dry matter content were also significantly correlated with disease index, with coefficients ranging from −0.685 to −0.968 and −0.527 to −0.962, respectively. As the disease index increased, these three indices gradually decreased. There were significant differences in weight of individual fruit, soluble solids content, and dry matter content among different disease levels (Table 3), and they gradually decreased with the epidemic of kiwifruit brown spot.

### 3.3. Effects of Disease Severity on Loss of Kiwifruit

The linear regression between disease index and the weight of individual fruit provided a good fit. All models derived were statistically significant (*p* < 0.05) and usable for subsequent analysis (Table 4). The coefficients of the disease variable (=slope) of these models ranged from −1.567 to −0.125 (Table 4), indicating the amount of weight of individual fruit loss per unit increase in disease index. The intercept of these models determined the weight of individual fruit with no disease occurrence. The yield loss was calculated for each region in different years with an increasing disease index. The yield loss of the Lushan region in 2020 was the highest (62.531%), with that of the Lushan region in 2019 being the lowest (8.271%). The Dujiangyan region in 2021 and the Anzhou region in 2019 were lower, less than 15%, and others ranged from 15.919% to 30.015%, with an average yield loss of 22.652%. These results indicated that kiwifruit brown spot significantly reduced yield by decreasing the weight of individual fruit. Meanwhile, the grade of kiwifruit was determined by the weight of individual fruit. Because the price of kiwifruit is directly influenced by the grade, the economic impact of downgrading tended to be greater than that of yield loss. In this study, most kiwifruits were downgraded one grade due to kiwifruit brown spot, with the exception of Lushan region in 2020, which was downgraded two grades, and Dujiangyan region in 2021, which received no downgrade.

### 3.4. Principal Component Analysis of Fruit Quality

Determining suitability for PCA depicted that the values of KMO and Bartlett’s tests were 0.615 and 110.726, respectively. When the degree of freedom is 21, the significance level has been reached, and the null hypothesis can be rejected, and *p* = 0.000 < 0.05, which indicated that the variables were not independent, so the data are appropriate for PCA. As shown in Table 5, the results demonstrated that the cumulative of the first three principal components (eigenvalue > 1.0) reached 88.139%, meaning that the information contained in the three principal components accounted for 88.139% of the total information, which could be used for subsequent analysis. According to the eigenvector, the first principal component (PC1) was mainly determined by dry matter content, soluble solids content, and the weight of individual fruit, indicating that the intrinsic flavor quality of kiwifruit was especially important in quality assessment. The contribution of PC2 was 18.316%, and the primary quality indices were firmness, shape index, and shelf life. PC3 was mainly determined by the disease incidence of fruit.

### 3.5. Comprehensive Evaluation of Fruit Quality

In this study, the scores of each plot were calculated based on the eigenvectors of each principal component, and then the contribution was used as the weight to calculate the cumulative sum of the product of the scores of the first three principal components of each plot and their respective weights to obtain the comprehensive score Dn. According to Figure 3, the comprehensive score of the Dujiangyan region in 2020 was the highest, whereas that of the Lushan region in 2021 was the lowest. There were no significant differences in comprehensive score of the same region in different year. The fruit quality of kiwifruit was the highest in 2020, followed by 2019, and was the worst in 2021 (Figure 4). There were significant differences in the comprehensive scores among different disease levels, and fruit quality was significantly reduced with the occurrence and epidemic of kiwifruit brown spot.

### 3.6. Effect of Kiwifruit Brown Spot on Fruit Quality

To clarify the effect of kiwifruit brown spot on fruit quality, the correlation between disease index and comprehensive score of each plot was determined. The Pearson correlation coefficient was −0.762 (*p* < 0.01), indicating that there was a significant correlation between them. Measured by annual data, the Pearson correlation coefficient was −0.844 (*p* < 0.01) in 2019, −0.807 (*p* < 0.01) in 2020, and −0.864 (*p* < 0.01) in 2021. The comprehensive score represented fruit quality. The above showed that the occurrence and epidemic of kiwifruit brown spot would seriously affect kiwifruit quality.

Integrated pest management (IPM) does not require the complete elimination of harmful organisms, but rather aims to control the population of harmful organisms below the economic injury level. To determine the damage threshold of kiwifruit brown spot, the disease loss model was constructed. The Boltzmann model was developed based on the disease index and comprehensive score of each plot (Figure 5). The residuals of the sample were closely distributed on both sides of the trend line, indicating that the model was accurate (Figure 6). The corresponding equation is as follows.
$$y=0.34458+0.34781/(1+\exp\frac{x-81.61920}{20.67692})$$

The model is an inverted sigmoidal shape curve. As the disease index increased, the comprehensive score decreased slowly, which had little impact on fruit quality at this stage. Once the inflection point was reached, the comprehensive score decreased rapidly as the disease index increased. When the disease reached a certain point of development, the impact on fruit quality essentially leveled off. We considered it harmless when the loss rate was below 5%. Using the above formula, the damage threshold of kiwifruit brown spot was calculated to be 36.14. In other words, if the disease index is controlled below 40, the impact of kiwifruit brown spot on fruit quality is negligible.

## 4. Discussion

In this study, the shape index, weight of individual fruit, firmness, soluble solids content, dry matter content, shelf life, and disease incidence of fruit were determined. Weight of individual fruit relates to yield and grade, soluble solids content and dry matter content are significant inner-quality indices, shape index and firmness are significant indices of sensory quality, and shelf life and disease incidence of fruit are significant postharvest indices in kiwifruit. These indices could provide a comprehensive evaluation of the kiwifruit quality. In previous research, the soluble solids content was typically determined by calculating the average value after fruit softening. However, we observed that soluble solids content could be affected by firmness, and that the method for calculating the average value could result in errors due to varying sample firmness. Therefore, modeling improved the accuracy of the soluble solids content, which was obtained by using the uniformly edible standard’s firmness value. In addition, the correlation between the disease index and each quality index was analyzed. Similarly to the research conclusion of Wang et al. [21] and Wu et al. [22], as the disease index increased, the weight of individual fruit, soluble solids content, and dry matter content decreased gradually.

To quantify yield losses caused by plant disease, a range of disease damage levels must be established. In this study, varying levels of disease severity were selected at each site, and linear regression was used to determine the theoretical value, which can be used to intuitively assess the adverse impact of the disease. Meanwhile, there were differences in site conditions of the different regions. Covering diverse ecological conditions, the impact of kiwifruit brown spot on yield and fruit quality was more effectively demonstrated. The estimation of disease loss is the study of the correlation between the occurrence or epidemic of the disease and the resulting decrease in yield and quality, which is categorized primarily as yield losses, quality reduction, and complex loss type [26]. The loss caused by kiwifruit brown spot is a complex loss type, whose measurement is more complex and influenced by market factors, and causes direct economic losses. Due to kiwifruit brown spot, kiwifruits have been downgraded one grade in most sites. Based on the market purchase price of ‘Hongyang’ kiwifruit, the price decreased by 4 yuan/kg for each grade down. The average kiwifruit yield was 15,000 kg/ha, so the direct economic loss was 60,000 yuan/ha. In addition, kiwifruit brown spot caused yield losses, with an average yield loss of 22.652%, so the total loss was 73,591 yuan/ha.

The impact of kiwifruit brown spot on quality between different experimental sites using raw data of fruit indices could not be comparatively analyzed directly. Therefore, to better reveal the effects of the disease on kiwifruit quality, a comprehensive evaluation was carried out by principal component analysis in this study. Under the premise of no or little loss of the original information, the original indices with a large number of values that were correlated with one another were transformed into new comprehensive indices with a small number of values that were either independent or correlated with one another, thereby avoiding the interference of repeated information. The fruit-quality index measurement units vary, and the data dimensions are inconsistent. Therefore, the raw data should be standardized. Considering the positive and negative effects of different indices on fruit quality, the membership function method was used to transform the raw data to make the results more objective and reasonable. Liao et al. [27] has utilized the membership function method to conduct a comprehensive evaluation and analysis of the fruit quality pollinated with different pollen donors. Cozzolino et al. [28] comparatively analyzed volatile metabolites, quality, and the sensory attributes of Actinidia chinensis based on PCA.

The first principal component was mainly determined by dry matter content, soluble solids content, and weight of individual fruit, which were significantly correlated with disease index. This indicated that these three indices largely determined the impact of kiwifruit brown spot on fruit quality. The saturation effect is the primary reason why there is little effect below the damage threshold. In other words, a portion of the photosynthetic area of a healthy plant belongs to the redundant photosynthetic area, and when the occurrence of disease is not serious, the actual harm is in this portion. Meanwhile, our research found that if the severity of kiwifruit brown spot was below 3 (i.e., the disease index below 33.33), the defoliation rate was very low and the effect on photosynthesis was minimal (unpublished). Therefore, the damage threshold was 36.14, which was reliable and in accordance with the practical production. Our future work will focus on controlling disease index below the damage threshold under different meteorological conditions.

## 5. Conclusions

Our findings showed that kiwifruit brown spot caused the average yield loss at the rate of 22.652%, leading to the fruit quality being downgraded by one grade. The damage threshold for kiwifruit brown spot based on the disease loss model was calculated to be 36.14 (disease index), which could provide good guidance for field prevention and control. This study is the first report on the effects of fungal disease on the yield and fruit quality of kiwifruit under natural conditions, which was an effective method of revealing the adverse impact of leaf-spot disease on yield and quality.

## Figures and Tables

**Figure 1 jof-11-00593-f001:**
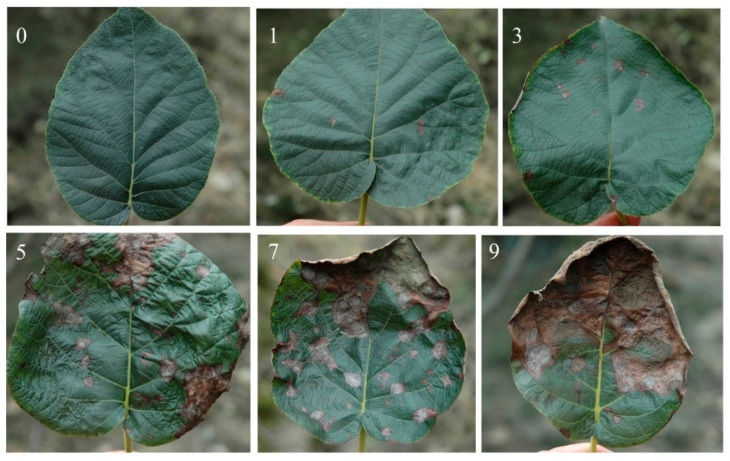
The standard of classification for kiwifruit brown spot using the severity grading of 0, 1, 3, 5, 7, and 9, defined as follows: 0 = no visible lesion; 1 = disease spot of leaf area <5%; 3 = 5 to 25%; 5 = 26 to 50%; 7 = 51 to 75%; 9 = more than 75% and defoliation.

**Figure 2 jof-11-00593-f002:**
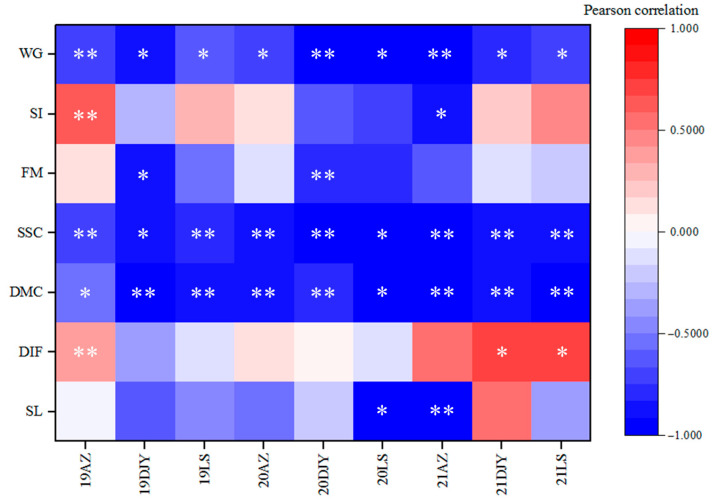
The correlation analysis between disease index and each fruit-quality index in trials over three years and three regions. WG: weight of individual fruit; SI: shape index; FM: firmness; SSC: soluble solids content; DMC: dry matter content; DIF: disease incidence of fruit rot; SL: shelf life. AZ: Anzhou region; DJY: Dujiangyan region; LS: Lushan region. Data are analyzed by Pearson correlation (* and ** indicate *p* < 0.05 and 0.01, respectively).

**Figure 3 jof-11-00593-f003:**
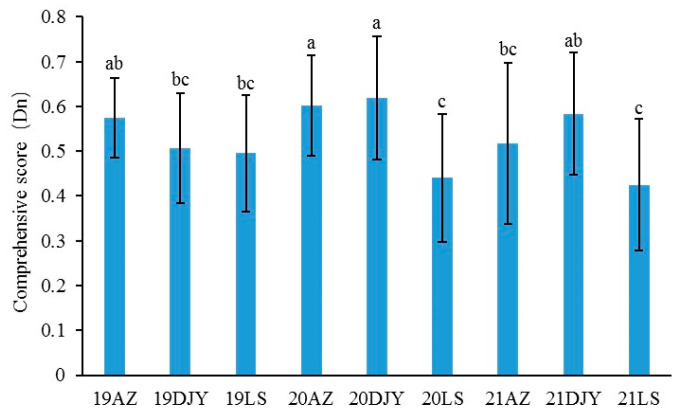
Comprehensive evaluation of fruit quality in three regions and three years. AZ: the Anzhou region; DJY: the Dujiangyan region; LS: the Lushan region. The letters on the columns indicate significant differences (*p* < 0.05) within each site according to Duncan’s new multiple-range test. Error bars represent standard error.

**Figure 4 jof-11-00593-f004:**
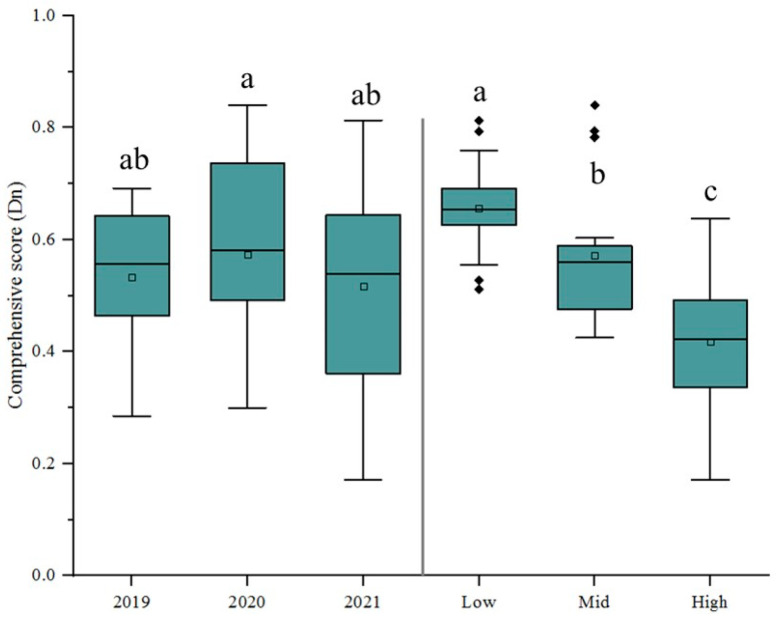
A comprehensive evaluation of fruit quality in trials over three years and different disease levels with 0, 1, 3 fungicide applications. The small square is the average value; the thick black line is the median; the whiskers extend to minimum and maximum values; and the dots are the outliers. The letters indicate significant differences at *p* < 0.05 determined by one-way ANOVA followed by Duncan’s new multiple-range test.

**Figure 5 jof-11-00593-f005:**
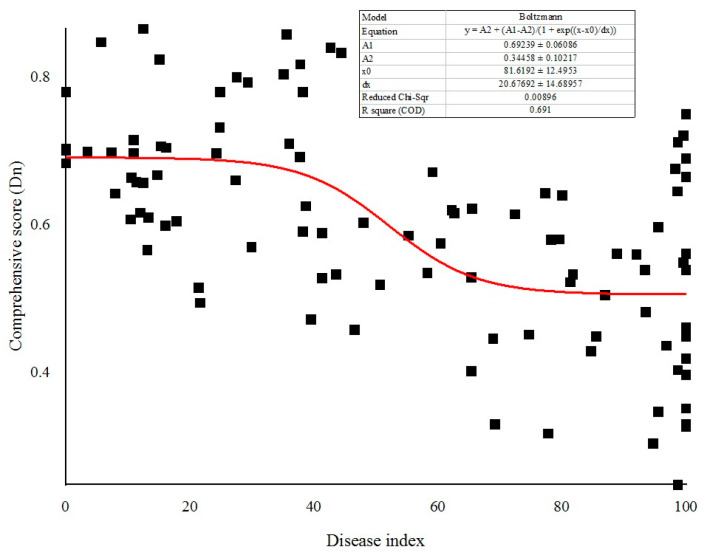
Disease loss model according to disease index and comprehensive score. Values for the parameters and fit statistics of the Boltzmann model used to describe this relationship are provided.

**Figure 6 jof-11-00593-f006:**
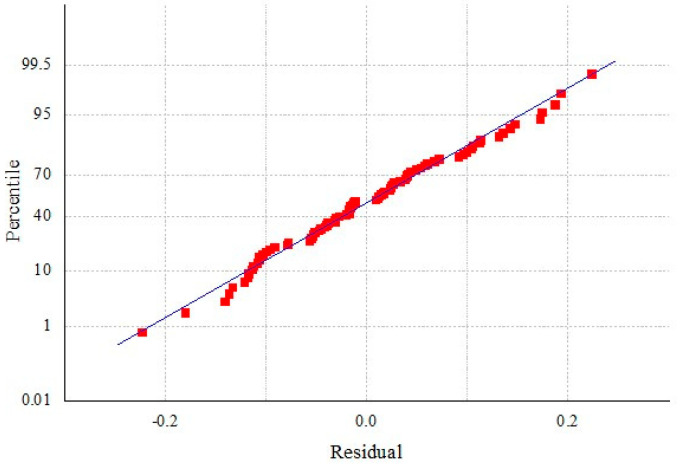
The sample residuals of the Boltzmann disease loss model were distributed symmetrically around the trend line for each plot.

**Table 1 jof-11-00593-t001:** The results of the screening model for common mathematical functions. The optimal-function model parameter values and fit statistics.

Function	Model	R^2^	F Value
Linear regression	Linear	0.75105	10,682.68
Convolution	Voigt	0.76479	2876.04
Exponential	LangevinMod	0.76783	3901.38
Growth	Boltzmann	0.77699	32,802.58
Logarithm	Log3P1	0.75969	5595.63
Peck Functions	InvsPoly	0.76852	1956.59
Piecewise	PWL2	0.76573	3855.89
Polynomial	Poly	0.77222	1330.84
Power	Asym2Sig	0.76699	2328.48
Rational	Holliday1	0.76168	42,748.51
Waveform	Sine	0.76389	3816.55
Chromatography	Gauss	0.76374	3813.39
Enzyme Kinetics	SubstrateInhib	0.75631	41,780.65
Statistics	Logistic	0.76660	32,746.11

**Table 2 jof-11-00593-t002:** Statistical characters of disease index and fruit-quality indices in trials over three years.

Indices	*n*	Min	Max	Mean	SD	Skewness	Kurtosis	CV (%)
DI	81	0.00	100.00	53.28	35.36	–0.012	–1.557	66.37
WIF (g)	81	57.30	177.30	94.18	20.89	1.060	2.097	22.04
SI	81	0.969	1.225	1.102	0.068	–0.323	–0.716	6.11
FM (kg/cm^2^)	81	3.727	7.830	5.550	0.826	0.348	–0.230	14.78
SSC (%)	81	12.80	21.20	16.77	1.89	0.169	–0.556	11.22
DMC (%)	81	15.12	25.12	19.13	1.81	0.577	1.020	9.39
DIF (%)	81	0.00	35.00	3.64	7.25	2.613	7.163	197.73
SL (d)	81	5.00	31.00	15.49	6.88	0.674	–0.635	44.13

Note: DI: disease index; WIF: weight of individual fruit; SI: shape index; FM: firmness, SSC: soluble solids content; DMC: dry matter content; DIF: disease incidence of fruit rot; SL: shelf life.

**Table 3 jof-11-00593-t003:** The effect of three disease levels with 0, 1, and 3 fungicide applications on the characteristics of fruit-quality indices.

	Low Disease Level	Medium Disease Level	High Disease Level
DI	14.24 ± 9.14 c	49.01 ± 12.10 b	89.87 ± 10.61 a
WIF (g)	108.57 ± 22.06 a	95.66 ± 14.39 b	80.73 ± 12.39 c
SI	1.095 ± 0.070 a	1.115 ± 0.043 a	1.103 ± 0.076 a
FM (kg/cm^2^)	5.688 ± 0.749 a	5.518 ± 0.778 a	5.444 ± 0.914 a
SSC (%)	18.02 ± 1.60 a	16.25 ± 1.19 b	15.92 ± 1.86 c
DMC (%)	20.16 ± 1.80 a	19.19 ± 1.13 b	18.19 ± 1.60 c
DIF (%)	1.83 ± 4.30 b	2.35 ± 5.00 b	5.88 ± 9.50 a
SL (d)	15.23 ± 5.81 b	17.76 ± 8.59 a	14.59 ± 6.76 b

Note: DI: disease index; WIF: weight of individual fruit; SI: shape index; FM: firmness; SSC: soluble solids content; DMC: dry matter content; DIF: disease incidence of fruit rot; SL: shelf life. Number of samples: low (*n* = 27), middle (*n* = 24), high (*n* = 30). Mean values followed by different letters in same line are significantly different (*p* < 0.05) according to Duncan’s new multiple-range test.

**Table 4 jof-11-00593-t004:** Significance levels from linear regression analysis of the effect on yield loss for each region in different years.

Year	Region	Intercept	Slope	R^2^	*p* Value	AWIF	YL (%)	TG	AG
2019	Anzhou	137.259	−0.502	0.554	0.001	119.279	13.099	I	II
	Dujiangyan	119.750	−0.525	0.766	0.022	97.833	18.302	II	III
	Lushan	83.207	−0.125	0.425	0.022	76.325	8.271	III	IV
2020	Anzhou	115.101	−0.336	0.527	0.027	96.778	15.919	II	III
	Dujiangyan	144.633	−0.713	0.964	0.000	101.222	30.015	I	II
	Lushan	235.072	−1.567	0.880	0.018	88.080	62.531	I	III
2021	Anzhou	119.218	−0.716	0.976	0.000	85.150	28.576	II	III
	Dujiangyan	95.042	−0.213	0.685	0.003	83.720	11.913	III	III
	Lushan	100.361	−0.225	0.498	0.034	85.067	15.239	II	III

Note: AWIF: actual weight of individual fruit; YL: yield loss; TG: theoretical grade; AG: actual grade.

**Table 5 jof-11-00593-t005:** Principal component analysis of fruit-quality indices. The eigenvector, eigenvalue, contribution rate and cumulative contribution of seven principal components.

	PC1	PC2	PC3	PC4	PC5	PC6	PC7
WIF	0.732	0.094	0.326	−0.355	0.128	−0.437	−0.121
SI	−0.392	0.550	0.532	−0.373	0.215	0.250	0.112
FM	−0.041	0.811	0.116	0.508	−0.186	−0.114	−0.149
SSC	0.789	0.102	0.118	−0.021	−0.484	0.064	0.339
DMC	0.854	0.043	−0.063	−0.098	−0.025	0.395	−0.315
DIF	0.333	−0.412	0.609	0.507	0.287	0.070	0.054
SL	0.554	0.362	−0.484	0.118	0.524	0.004	0.198
Eigenvalue	2.461	1.282	1.026	0.805	0.689	0.432	0.306
Contribution (%)	55.160	18.316	14.663	4.493	3.836	2.167	1.365
Cumulative (%)	55.160	73.476	88.139	92.632	96.468	98.635	100.000

Note: WIF: weight of individual fruit; SI: shape index; FM: firmness; SSC: soluble solids content; DMC: dry matter content; DIF: disease incidence of fruit rot; SL: shelf life.

## Data Availability

The data presented in this study are available on request from the corresponding author.

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
