# Peer review of "Different Disease Levels Reveal Kiwifruit Brown Spot Impacts on Fruit Yield and Quality"

_jof, 2025, doi:10.3390/jof11080593_

Round 1

Reviewer 1 Report

This study is well-designed; it has covered all the necessary parameters to determine the influence of disease on production quality and yield. It has taken into account several locations and various disease levels. The contribution of this study is evident. It only needs to be more accessible to the public, less specialized in statistics and principal components. This often occurs when the authors are highly specialized individuals who handle these concepts extensively and have little contact with professionals in other fields. But it's not insurmountable; it simply makes the study a little more understandable for the general research public.

I see the text as correct, although I am curious (point 2.2 of materials and methods) as to why the authors have used the severity grading of 1, 3, 5, 7, and 9, and not 1 to 5. I know that they have applied another author's grading, but why is it not correlated?. 

In Results, I would delete the sentence on line 198 since it is a conclusion and start directly with "The result indicated a significant correlation (Figure 3).....". The same with the sentence on line 209, it is a conclusion, I would start with the second sentence where it begins to show the results.
In section 3.5 of the results, the authors say "There were significant differences in the comprehensive score of the same year in different regions, and within the same region in different years. The fruit quality of kiwifruit in 2020 was the highest, followed by 2019 and the worst in 2021 (Figure 5)." Well, I go to see Figure 5 and see the one way ANOVA statistics as 2019 (ab), 2020(a), and 2021 (ab).

A one-way ANOVA was conducted to examine differences between the years 2019, 2020, and 2021. The post-hoc test revealed that the group means for 2019, 2020, and 2021 were not significantly different from each other. This is indicated by the shared grouping letters in the post-hoc results: 2019 (ab), 2020 (a), and 2021 (ab). Since all year pairs share at least one letter in common, it can be concluded that there are no statistically significant differences in the means between any of the three years.

Figure 2 is very small; you can barely make out the values on the axes, and it only serves to indicate the normal distribution of the data. I would move the figure to the supplementary section.

I don't understand Figure 3 as it stands. The colors are very dark and the asterisks are indistinguishable. P<0.05 and P<0.01 are mixed together. It would be better to see them all in a single P.
Table 2 doesn't make sense to me; I find Table 3 more useful. Finally, one observation: whenever I've seen a principal components analysis in an article, I've always seen it represented on axes, either in 2D or 3D. I don't understand why the authors present it in Table 5.

In general, I would do a good review of figures and tables to make the study more comprehensive.

Author Response

Response for the reviewer 1

Dear Reviewer,

On behalf of all our coauthors, we thank you very much for the positive evaluation and informative revision of our manuscript, which are very important to us. Following your advice, we have carefully addressed these issues as follows:

Q1: I see the text as correct, although I am curious (point 2.2 of materials and methods) as to why the authors have used the severity grading of 1, 3, 5, 7, and 9, and not 1 to 5. I know that they have applied another author's grading, but why is it not correlated?

Response: Thanks for your comments. We believe that both grading systems were correct, and the severity grading of 1, 3, 5, 7, and 9 was widely used in the leaf spot disease, which has no impact on calculating disease index.

Q2: In Results, I would delete the sentence on line 198 since it is a conclusion and start directly with "The result indicated a significant correlation (Figure 3).....". The same with the sentence on line 209, it is a conclusion, I would start with the second sentence where it begins to show the results.

Response: Thanks for your advice. We have revised these descriptions according to your suggestion, the details were shown in revised manuscript.

Q3: In section 3.5 of the results, the authors say "There were significant differences in the comprehensive score of the same year in different regions, and within the same region in different years. The fruit quality of kiwifruit in 2020 was the highest, followed by 2019 and the worst in 2021 (Figure 5)." Well, I go to see Figure 5 and see the one way ANOVA statistics as 2019 (ab), 2020(a), and 2021 (ab).

A one-way ANOVA was conducted to examine differences between the years 2019, 2020, and 2021. The post-hoc test revealed that the group means for 2019, 2020, and 2021 were not significantly different from each other. This is indicated by the shared grouping letters in the post-hoc results: 2019 (ab), 2020 (a), and 2021 (ab). Since all year pairs share at least one letter in common, it can be concluded that there are no statistically significant differences in the means between any of the three years.

Response: Your advice looks helpful for us. We have carefully examined the statistics, and have revised incorrect descriptions. The details were shown in revised manuscript.

Q4: Figure 2 is very small; you can barely make out the values on the axes, and it only serves to indicate the normal distribution of the data. I would move the figure to the supplementary section.

Response: Thanks for point this out. We have moved figure 2 to the supplementary section.

Q5: I don't understand Figure 3 as it stands. The colors are very dark and the asterisks are indistinguishable. P<0.05 and P<0.01 are mixed together. It would be better to see them all in a single P.

Response: Thanks for point this out. We have changed the color of the asterisk to white for easier distinction. The mixing of P<0.05 and P<0.01 was to better distinguish significant differences in fruit quality indices across different regions.

Q6: Table 2 doesn't make sense to me; I find Table 3 more useful. Finally, one observation: whenever I've seen a principal components analysis in an article, I've always seen it represented on axes, either in 2D or 3D. I don't understand why the authors present it in Table 5.

Response: Thanks for your comments. Table 2 presents the central tendency, dispersion degree, and distribution pattern of the data through statistical characters and visualization methods, providing a foundation for subsequent analysis.

We believed that the results of principal components analysis could be presented in both tables and figures, and the choice depends on the purpose and requirements. In our research, tables might be a better choice for accurate comparisons and finding eigenvalue among numerous variables and principal components.

Reference: Chen, M.Y.; Zhao, T.T.; Liu, X.L.; Han, F.; Zhang, P.; Zhong, C.H. Factor analysis and comprehensive evaluation of fruit quality of ‘Jinyan’ kiwifruit. Plant. Sci. J. 2021, 39, 85-92.

Reviewer 2 Report

Major Comments:

The submitted paper to JoF “Different Disease Levels Reveal Kiwifruit Brown Spot Impacts on Fruit Yield and Quality" by Yuhang Zhu et al. is interesting and it is well constructed. It is a fair contribution to know more about the Kiwifruit Brown Spot disease and how it impacts on kiwi fruit productivity and economics in China.

The paper is clear, well written and well organised.

The tittle is ok to me.

The abstract is clear and embodies well the article, pointing out the main contributions of the paper for the fungal disease in question and its impact on kiwi fruit productivity.

Keywords were generally well chosen, but I wouldn’t repeat those that are already present in the tittle.

The Introduction of the paper is well written and well organized, and it is a good state of the art.

The objectives of the study are plainly indicated at the end of introduction.

The Materials and Methods section is well organised and the subsections created are useful for readers to following the paper; the methods are adequately described and with some detail.

Moreover, the authors employed adequate statistics.

Results are also well organised and properly described, with adequate figures and tables. Figures are of good quality, but the legends need to be improved. The same happens with most of the tables’ tittles. In addition, I would prefer that the figures and tables would be placed along the text and not at the end to better understand and follow the paper.

Discussion is well constructed and articulated, covering all points of the “Results” section, and it is well supported by the scientific literature.

The section “Conclusions” could be more developed (maybe one or two paragraphs more would be good!). I would rephase them. Also, the first sentence does not make too much sense to me. Please, see my detailed comments.

The list of References is fair and enough, but please do revise the format of the references across the text, by numbers as it is required by the journal.

In summary, this is a good article, in my opinion, that can deserve publication in “Journal of Fungi” .

Detailed comments:

Introduction:

Lines 43-44 in Introduction: the reader does not know what disease the authors are referring to. Is it the fungal disease? The bacterial one? Are the authors referring to both? Please be clear!

Line 52: the name of the genus Alternaria must be in italics.

Materials and Methods:

Line 99: I would explain better and detailed more the “out of control fungicide effects”.

Line 135: please correct the typo/gap…

I would transform lines 144-146 into a table. But this is not mandatory to me…

Results:

In lines 260-261, I would rephrase the sentences. As they are written, it turns out a little bit confusing…

Legend of figure 2 needs to be completed. The parameters of the 8 graphs need to be explained and listed in the legend (DI; WIF; SI; …). And the abbreviations need also to be listed in the text in section 3.2. Figures must be self-informative.

The same happens in Figure 1. Legends need to be improved and developed to be complete and totally informative legends. The same problem about the other figures’ legends. These must be better explained in order to the figures become self-informative.

Also, do the same with the tables’ tittles (for tables 3 and 5, as the authors did for tables 2 and 4).

Discussion:

Sentences in lines 334 and 335 needs to be rephrased.

Conclusions:

Conclusions need to be developed and rephrased. In addition, the first paragraph of the conclusions should be rephrased to be consistent to the study, in my opinion. I mean, the first sentence is too general for me and not particularly related to this study… The paper does not deal so much with the application of fungicides and its effect on the spread of the disease or in fruit production and quality. I mean, as far as I understand, the fungicide application is the same in every plot. Moreover, the plots where the out of control fungicide effects occurred were discharged from the analysis.

Author Response

Response for the reviewer 2

Dear Reviewer,

On behalf of all our coauthors, we thank you very much for the positive evaluation and informative revision of our manuscript, which are very important to us. Based on your suggestions, we have carefully considered all comments and we have tried our best to revise our manuscript. You can find the detailed responses below.

Q1: Keywords were generally well chosen, but I wouldn’t repeat those that are already present in the tittle.

Response: Thanks for point this out. We have revised 'kiwifruit brown spot' to 'Corynespora cassiicola' in the new submission.

Q2: Lines 43-44 in Introduction: the reader does not know what disease the authors are referring to. Is it the fungal disease? The bacterial one? Are the authors referring to both? Please be clear!

Response: Thanks for point this out. We have clear referred it to 'kiwifruit brown spot ' according to your suggestion.

Q3: Line 52: the name of the genus Alternaria must be in italics.

Response: Sorry for the mistake. We have corrected it.

Q4: Line 99: I would explain better and detailed more the “out of control fungicide effects”.

Response: Thanks for your advice. We have better explained“out of control fungicide effects”, the details were shown in revised manuscript.

Q5: Line 135: please correct the typo/gap…

Response: Sorry for the mistake. We have corrected it.

Q6: I would transform lines 144-146 into a table. But this is not mandatory to me…

Response: Thanks for your advice. We remain unchanged for format.

Q7: In lines 260-261, I would rephrase the sentences. As they are written, it turns out a little bit confusing…

Response: Your advice looks helpful for us. We have revised this sentences, the details were shown in revised manuscript.

Q8: Legend of figure 2 needs to be completed. The parameters of the 8 graphs need to be explained and listed in the legend (DI; WIF; SI; …). And the abbreviations need also to be listed in the text in section 3.2. Figures must be self-informative.

The same happens in Figure 1. Legends need to be improved and developed to be complete and totally informative legends. The same problem about the other figures’ legends. These must be better explained in order to the figures become self-informative.

Also, do the same with the tables’ tittles (for tables 3 and 5, as the authors did for tables 2 and 4).

Response: Thanks for your comments. We have completed Legend of figure 2 (now Figure S1) and explained the parameters of the 8 graphs in the legend. We have moved figure 2 to the supplementary section according to another reviewers’ suggestions. We have also listed the abbreviations in the text in section 3.2.

We have improved other figures’ legends and tables’ tittles to make them become self-informative.

Q9: Sentences in lines 334 and 335 needs to be rephrased.

Response: Thanks for your comments. We have revised this sentences, the details were shown in revised manuscript.

Q10: Conclusions need to be developed and rephrased. In addition, the first paragraph of the conclusions should be rephrased to be consistent to the study, in my opinion. I mean, the first sentence is too general for me and not particularly related to this study… The paper does not deal so much with the application of fungicides and its effect on the spread of the disease or in fruit production and quality. I mean, as far as I understand, the fungicide application is the same in every plot. Moreover, the plots where the out of control fungicide effects occurred were discharged from the analysis.

Response: Thanks for your advice. We have revised the section “Conclusions”, the details were shown in revised manuscript.

Round 2

Reviewer 1 Report

The authors have answered the questions posed to them, helping me better understand their work. They have also implemented the suggestions, and now the work looks more complete. I believe the authors' contribution to this area of knowledge is positive.

All improvements are correct

Author Response

Dear reviewer 1,

Thanks very much for your recognition on publication our paper. On behalf of all our coauthors, we would like to express our great appreciation to you.

Yours sincerely,

Guoshu Gong, PhD

Reviewer 2 Report

The revised version of the paper “Different Disease Levels Reveal Kiwifruit Brown Spot Impacts on Fruit Yield and Quality" by Yuhang Zhu et al. was improved, and in my opinion, it can now be accepted for publication in JoF. I emphasize again that it is a good contribution to the knowledge of the Kiwifruit Brown Spot disease and how it impacts on kiwi fruit productivity and kiwi fruit economics in China.

Almost my major issues and questions were properly addressed by the authors.

Moreover, Conclusions were also improved.

But, I just wonder again why the figures and tables are not inserted along the text, and only at the end of the manuscript, as I pointed out in my report 1, in the major comments concerning the Results section…

The legends’ figures and tables’ tittles were improved to become self-informative. But I would improve once more the tittle of Table 1 and the legends of figures 5 and 6, linking the two sentences in each fugure legend and tittle of the table.

Author Response

Dear reviewer 2,

Thanks very much for your recognition on publication our paper. On behalf of all our coauthors, we are grateful for your effort reviewing our paper and positive feedback. Additionally, we are very sorry for missing your comments in report 1, and figures and tables have been inserted along the text in new submission according to your advice. We have improved once more the tittle of table 1 and the legends of figures 5 and 6.

Yours sincerely,

Guoshu Gong, PhD